# Predictive Factors Involved in Postpartum Regressions of Cytological/Histological Cervical High-Grade Dysplasia Diagnosed during Pregnancy

**DOI:** 10.3390/jcm10225319

**Published:** 2021-11-15

**Authors:** Yvan Gomez, Vincent Balaya, Karine Lepigeon, Patrice Mathevet, Martine Jacot-Guillarmod

**Affiliations:** 1Colposcopy Unit, Women-Mother-Child Department, Lausanne University Hospital, 1011 Lausanne, Switzerland; v.balaya@hopital-foch.com (V.B.); karine.lepigeon@chuv.ch (K.L.); patrice.mathevet@chuv.ch (P.M.); Martine.jacot-guillarmod@chuv.ch (M.J.-G.); 2Faculty of Biology and Medicine, University of Lausanne, 1015 Lausanne, Switzerland; 3Department of Gynecology and Obstetrics, Foch Hospital, 92150 Suresnes, France

**Keywords:** high-grade dysplasia, cervical cancer, cervical intraepithelial neoplasia, CIN, HSIL, ASC-H, pregnancy

## Abstract

Objective: The aim of this study was to describe the evolution of high-grade cervical dysplasia during pregnancy and the postpartum period and to determine factors associated with dysplasia regression. Methods: Pregnant patients diagnosed with high-grade lesions were identified in our tertiary hospital center. High-grade lesions were defined either cytologically, by high squamous intraepithelial lesion/atypical squamous cells being unable to exclude HSIL (HSIL/ASC-H), or histologically, with cervical intraepithelial neoplasia (CIN) 2+ (all CIN 2 and CIN 3) during pregnancy. Postpartum regression was defined cytologically or histologically by at least a one-degree reduction in severity from the antepartum diagnosis. A logistic regression model was applied to determine independent predictive factors for high-grade cervical dysplasia regression after delivery. Results: Between January 2000 and October 2017, 79 patients fulfilled the inclusion criteria and were analyzed. High-grade cervical lesions were diagnosed by cytology in 87% of cases (69/79) and confirmed by histology in 45% of those (31/69). The overall regression rate in our cohort was 43% (34/79). Univariate analysis revealed that parity (*p* = 0.04), diabetes (*p* = 0.04) and third trimester cytology (*p* = 0.009) were associated with dysplasia regression. Nulliparity (OR = 4.35; 95%CI = (1.03–18.42); *p*= 0.046) was identified by multivariate analysis as an independent predictive factor of high-grade dysplasia regression. The presence of HSIL on third-trimester cervical cytology (OR = 0.17; 95%CI = (0.04–0.72); *p* = 0.016) was identified as an independent predictive factor of high-grade dysplasia persistence at postpartum. Conclusion: Our regression rate was high, at 43%, for high-grade cervical lesions postpartum. Parity status may have an impact on dysplasia regression during pregnancy. A cervical cytology should be performed at the third trimester to identify patients at risk of CIN persistence after delivery. However, larger cohorts are required to confirm these results.

## 1. Introduction

In current obstetrical practice, antenatal consultations are commonly considered as an opportunity to screen for cervical cancer. High-grade dysplasia is most frequently diagnosed during the childbearing years, with an incidence of 8.1/1000 women aged 25–29 years [1]. Cervical intraepithelial neoplasia (CIN) is diagnosed in 1–7% of pregnant women [2,3], among whom the prevalence of high-grade lesions (defined as CIN2+) is 0.5% [4]. The progression of high-grade lesions during pregnancy to microinvasive lesions ranges from 0% to 13% [5,6]. Human papilloma virus (HPV) infections appear to be more frequent among pregnant women, which may be related to the immunotolerance observed in pregnancy [7,8].

According to international criteria [9], pregnant patients with cytological abnormalities should be investigated by colposcopy in order to exclude a cancer by targeted biopsy. Although assessment of the cervix in pregnancy may be complicated due to pelvic congestion, colposcopic criteria remain the same as for non-pregnant women. In addition, the transformation zone is well visualized due to the eversion of the endocervix as pregnancy progresses. Cytology, colposcopy, and targeted biopsies are as reliable during pregnancy as for non-pregnant-women [10].

As regards the evolution of cervical dysplasia during pregnancy, the literature has shown a trend toward increased postpartum regression [11]. This regression is possibly related either to the return of the immune system or to a stimulation of it. A previous article suggested that during vaginal delivery, the cervical microlesions induced inflammation and, consequently, regression of the lesions. However, these results remain controversial. 

Through a unicentric cohort of pregnant women diagnosed with high-grade cervical dysplasia during pregnancy, the aim of this study was to describe the evolution of high-grade cervical dysplasia during pregnancy and the postpartum period and to determine the factors that can be associated with dysplasia regression.

## 2. Materials and Methods

We retrospectively reviewed the medical records of pregnant women with high-grade lesions who were referred to the colposcopy unit of our tertiary hospital between January 2000 and September 2017. High-grade lesions were defined either cytologically by high squamous intraepithelial lesion/atypical squamous cells that cannot exclude HSIL (HSIL/ASC-H) or histologically with high-grade cervical intraepithelial neoplasia (CIN 2 and CIN 3), during pregnancy. All CIN 2 and CIN 3 high-grade histologic dysplasia were classified as CIN2+. Other data abstracted from the medical record included parity, tobacco use, diabetes, presence of HPV, first/second/third trimester feature, delivery route and postpartum feature.

Patients were excluded in cases of incomplete cytological or histological ante/postpartum data about the evolution of dysplastic lesions or in cases of the presence of a factor that could affect the patient’s immunity (immunosuppressive treatment or acquired/innate immunodeficiency).

Cervical biopsies were not systematically performed but were required in cases of significant discordance between cervical cytology and colposcopic impression or those with colposcopic evidence of invasion. To ensure the expertise of the colposcopy assessment, all cases of high-grade lesions were supervised by a colposcopy expert (MJG), either by the image taken by the colpophotograph or directly during the colposcopy. The follow-up was based on a comparison with the images taken during the previous consultation.

When a diagnosis of high-grade dysplasia was confirmed in a pregnant woman, a quarterly follow-up was scheduled on the basis of typical obstetrical trimesters. At each visit, a cervical PAP smear and a colposcopy were performed, and an additional directed cervical biopsy was carried out in cases of suspected higher-grade lesions. The mode of delivery and other procedures were recorded as well.

In our cohort, the postpartum colposcopic follow-up was performed within six to eight weeks after childbirth. Postpartum regression was defined cytologically or histologically by a reduction of at least one-degree in severity from the antepartum diagnosis, such as HSIL regressed to LSIL/normal or CIN2+ regressed to CIN 1/normal. The histopathological findings in patients who underwent loop electrosurgical excision procedure (LEEP procedure) due to extensive lesions or cytology/colposcopic discordance were included to the analysis.

Qualitative data were compared by using the chi-square test and quantitative data by using the student *t*-test. Results are presented as 95% confidence intervals (95%CIs) or numbers (percentages). *p* values lower than 0.05 were retained as significant. Patients with CIN regression were compared to those with CIN persistence during the postpartum period. Relevant covariates associated with cervical dysplasia regressions that were significant (*p* < 0.05) in the univariate analysis were considered in a backward selection procedure to fit a multivariable model. A logistic regression model was applied to determine independent predictive factors for high-grade cervical dysplasia regression after delivery. All statistical analyses were carried out using XLStat Biomed software (AddInsoft V19.4, Paris, France). This study was approved by the ethical research committee of the canton de Vaud (25 July 2019, ID N 2017-01375).

## 3. Results

Ninety-four women diagnosed with high-grade dysplasia were identified. Among them, four patients with a cytological HSIL were excluded due to an extensive lesion during the antenatal period with colposcopic neoplasia suspicion, who subsequently underwent a LEEP procedure. Finally, only CIN 2+ lesions without invasive lesions were found in these four cases. No cases received corticosteroid treatment in our cohort; only one patient was known to have mother-to-child HIV infection with CD4+ in the normal range throughout her pregnancy. After exclusion of 15 patients, 79 patients fulfilled the inclusion criteria and were analyzed (Figure 1). The characteristics of the study population are presented in Table 1 and Table 2.

Patients had at least two antenatal examinations in 90% of cases (71/79) and all had a postpartum evaluation. High-grade lesions were diagnosed by cytology in 87% of cases (69/79), of which 45% (31/69) were histologically confirmed, whereas the remaining 13% (10/79) were found on biopsies that revealed CIN2+ after initial LSIL cytology. During colposcopic follow-up, no cases of invasive lesions were found. Delivery routes were vaginal in 68% of cases (54/79) and by C-section in 32% of cases (25/79). Among the patients who had a vaginal delivery, 27 delivered spontaneously, 14 were induced for an obstetric indication and 13 were unspecified.

At postpartum colposcopic follow up, all patients underwent colposcopy with cytology/histology except for three. These three cases underwent a LEEP procedure immediately postpartum due to CIN2+ extensive antenatal lesions, which was confirmed by the pathology. Of the 27 patients whose postpartum cytology revealed an HSIL, 18 (66.7%) histological samples confirmed a high-grade lesion (either by directed biopsy or conization result) and 6 (22%) samples did not. In the 34 cytological specimens showing an LSIL/ASCUS/normal lesion, only 20 (59%) histological specimens were collected, of which 10 (50%) confirmed a high-grade lesion. For patients who only underwent colposcopic evaluation (15/79), 13 histological samples were taken, 8 (62%) of which confirmed a HSIL lesion. Sixteen patients were only diagnosed cytologically and not histologically, of which only 3 (11%) were in the HSIL group. A total of 47 biopsies and 42 conizations were performed with a rate of 53% (25/47) and 79% (33/42), respectively, of persistent high-grade lesions. Seventy-one percent (56/79) of our cohort had a histological diagnosis at postpartum colposcopic follow up. The overall regression rate in our cohort was 43% (34/79).

Univariate analysis revealed that parity (*p* = 0.04), diabetes (*p* = 0.04) and third trimester cytology (*p* = 0.009) were associated with dysplasia regression. Age, smoking and delivery route did not impact on postpartum CIN regression rate (Table 3 and Table 4). By multivariate analysis, nulliparity (OR = 4.35; 95%CI= (1.03–18.42); *p*= 0.046) and presence of HSIL at third-trimester cervical cytology (OR = 0.17; 95%CI = (0.04–0.72); *p* = 0.016) were identified as independent predictive factors for dysplasia regression (Table 5).

## 4. Discussion

The aim of this study was to assess the clinical interest in knowing which factors would allow targeting patients at potential risk of maintaining a high-grade lesion or progressing to a cancerous lesion in the postpartum period and avoiding loss to follow up for adequate treatment. 

In our cohort, we reported a regression rate of high-grade lesion at 43% and no case of invasive lesions at postpartum colposcopy follow up. This rate is similar to those reported in the literature of 17–69% [4,5,12,13]. However, these studies show large disparities in terms of diagnostic criteria and therapeutic management. Considering all grades of dysplasia, caution should be paid to the interpretation of these data, since overall regression may design initial high-grade or low-grade lesions [6,11,14,15,16,17,18].

The correlation between cytology and final diagnosis within one degree of severity in pregnant woman has been found to be 78% [10], with cytology having an 88% positive predictive value for high-grade lesions [10]. Regarding colposcopic impressions, the results are consistent with final diagnoses in 73% of cases within one degree of severity [19]. In our cohort, we found a prenatal biopsy rate of 45% due to suspicion of cancerous lesion. This is due to overestimation of lesions related to colposcopic changes in pregnancy, which often leads to systematic sampling despite the examiner’s experience. This is described by Fader et al. [20], in their large study based on correlations between colposcopic impression and final diagnosis by colposcopy experts; out of 62 samples taken for colposcopic suspicion of CIN2+/neoplasia, only 55% confirmed a high grade lesion. The overestimation of the severity of lesions when performing colposcopies among pregnant women is the main reason for the non-concordance [19]. In our cohort, 34 of the 45 patients with diagnoses of persistent HSIL/CIN2+ at postpartum colposcopy follow up had a LEEP procedure confirming high-grade non-invasive dysplastic lesions.

In our univariate analysis, there was a statistically significant result for the variable gestational diabetes; however, this should be interpreted with caution due to the small patient cohort (eight patients). In contrast, our results support the idea that the parity status may have an impact on HSIL/CIN regression rate during pregnancy. This finding is concordant with the findings reported by Hong et al. [21]. The authors highlighted that the persistence of high-grade lesions was more frequent in multiparous patients (OR: 10.52; 95%CI: 1.36–81.01; *p* = 0.004) [21]. Compared to multiparous patients, nulliparous patients would also have longer exposure estrogen impregnation related to cervical ripening [22], as well as related proinflammatory cytokine signaling, leading to increased local vascularization and a recrudescence of immune cells such as myeloid-derived immune cells and lymphocyte cells. These cells may participate in inflammation and recovery processes [23]. The density of CD4+ T-lymphocytes was shown to increase by 4-fold in pregnant patients who are not in labor and by 10-fold in pregnant patients in labor [24].

There is some evidence that persistent, high-risk HPV infections in the postpartum period may be an underlying factor in patients with either persistent or progressing lesions [21]. Pregnancy leads to physiological changes that induce temporary immunomodulatory effects in downregulating the expression of inflammatory chemokines [25] increasing susceptibility to HPV infection. Hong et al. [21] showed that the persistence of high-grade lesions were more frequent in patients with persistent, high-grade HPV infections (OR = 5.25; 95%CI: 2.26–12.18; *p* < 0.001).

However, no differences were found between both groups in terms of presence of high-risk HPV. As for non-pregnant women, it would be interesting to investigate other markers, such as P16 and Ki67, and to assess their prognostic value for more accurately identifying patients at risk of developing invasive lesions. Nonetheless, there is no consensus on the use of P16 and Ki67 during pregnancy, and the influences of hormones and immunity on these proteins remain unknown. The literature is scarce with only one study that has been published; furthermore, it was based on a small series of cases [26].

Some authors raised the hypothesis that cervical desquamation related to vaginal delivery resulted in an inflammatory response that leads to increased regressions of cervical intraepithelial lesions in the postpartum period [11]. However, the role of the delivery route is still subject to debate and remains controversial. As suggested by our results, the route of delivery had no influence on HSIL/CIN 2+ regression rate. This result is concordant with that reported by Yost et al. [5], who reported no difference in regression rates according to mode of delivery in a prospective study of 153 histologically diagnosed high-grade dysplasia.

This study has several limitations. First, the retrospective design and the small sample size lead to insufficient power to extrapolate these results. Second, our study is mainly based on colposcopic and cytological criteria, histological samplings being carried out only when a lesion is suspected of being cancerous. Although cytology has a positive predictive value of 88% for high-grade lesions in pregnant women, histology remains the gold standard for the diagnosis of high-grade lesions, thus excluding high-grade cytological lesions lacking histological confirmation [4,5,6,12,14,15,21,27]. This implies that if the sample does not contain the high-grade lesion, this minimizes the risk and results in these patients not being monitored during pregnancy and postpartum [28]. However, the results of the current study underlined that cytological regression at the third trimester was predictive of HSIL/CIN regression after delivery. In addition, Ueda et al. reported the same observations as those underlined in our study, with a spontaneous regression in a quarter of their cohort in the second and third trimester before delivery. Persistent HPV infection is necessary for the development of precancers and cancers. Although our study does not report HPV typing, there is a ripening of the cervix due to hormonal change and, consequently, a return of the inflammatory system in the third trimester. We support the idea that this process allows clearance of the HPV infection and, consequently, a regression of the lesion. Cytology in the third trimester will allow this change to be observed and, therefore, predict the evolution of the lesions.

## 5. Conclusions

In conclusion, our study follows the ASCCP (American Society of Colposcopy and Cervical Pathology) guidelines. Although assessment of high-grade dysplastic lesions may be postponed in the postpartum period, our observations revealed that the clinical value of surveillance for high-grade dysplasia during pregnancy is to identify lesions with a potential risk of progression to cervical carcinoma in the postpartum period. Our results highlighted that parity status may have an impact on the regression of dysplasia during pregnancy. Cervical cytology should be performed in the third trimester to identify patients at risk of persistent CIN after delivery. Therefore, larger cohorts are required to confirm these results. A prospective study, including biomarkers such as P16 and Ki67, as well as immune cell characteristics/density and HPV typing would be relevant to gain more knowledge and increase the accuracy of diagnosis and management of high-grade cervical lesions during pregnancy and postpartum.

## Figures and Tables

**Figure 1 jcm-10-05319-f001:**
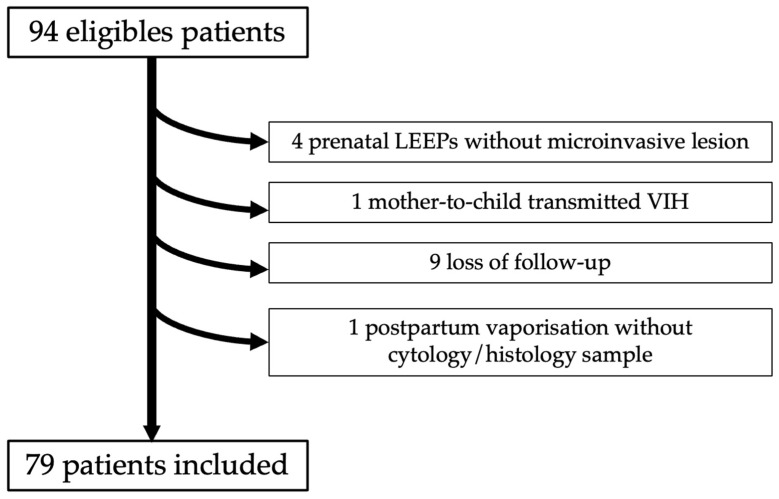
Flow-chart of population study.

**Table 1 jcm-10-05319-t001:** General patient characteristics.

Predictive Variable	Overall Population *n* = 79	
	*n* Mean ± SD	(%) (Range)
Age (years)		
Mean	29.7 ± 4.8	(20–41)
Parity status		
Nulliparity	40	53.3
Multiparity	35	46.7
Not specified	4	
Tobacco use		
Yes	23	30.7
No	52	69.3
Not specified	4	
Gestational diabetes		
Yes	8	10.7
No	67	89.3
Not specified	4	
Presence of HPV		
Yes	18	22.8
No	61	77.2
Delivery route		
Vaginal delivery	54	68.4
C-section	25	31.6

**Table 2 jcm-10-05319-t002:** Colposcopy-related patient characteristics by trimester.

Predictive Variable	First Trimester		Second Trimester		Third Trimester		Postpartum Colposcopic Follow Up	
	Overall Population *n* = 79							
	*n* Mean ± SD	(%) (Range)	*n* Mean ± SD	(%) (Range)	*n* Mean ± SD	(%) (range)	*n* Mean ± SD	(%) (Range)
Colposcopy performed								
Yes	68	86.1	71	89.9	60	76.0	76	96.2
No	11	13.9	8	10.1	19	24.0	3	3.8
Gestational age (weeks)								
Median	10	(1–20)	20	(12–29)	32	(24–39)		
Aspect								
HSIL	21	65.6	41	67.2	34	68.0	36	65.5
LSIL	10	31.3	15	24.6	14	28.0	12	21.8
Normal	1	3.1	5	8.2	2	4.0	7	12.7
Not specified	36		10		10		21	
Cytology								
HSIL	44	66.7	24	53.3	25	51.0	26	42.6
LSIL	14	21.2	13	28.9	16	32.6	10	16.4
ASC-H	8	12.1	3	6.7	8	16.3	1	1.6
Normal			5	11.1			24	39.3
Not performed	2		26		11		15	
Biopsy								
HSIL	19	90.5	17	81.0	9	69.2	25	53.2
LSIL	2	9.5	2	9.5	2	15.4	12	25.5
Normal			2	9.5	2	15.4	10	21.3
Not performed	47		50		47		29	

**Table 3 jcm-10-05319-t003:** Univariate analysis of general factors associated with postpartum CIN regression.

Predictive Variable	Persistence of CIN *n* = 45		Regression of CIN *n* = 34		*p*
	*n* Mean ± SD	(%) (Range)	*n* Mean ± SD	(%) (Range)	
**Age (years)**					
**Mean**	29.5 ± 4.8	(20–41)	30.1 ± 4.8	(21–41)	0.58
**Parity status**					
Nulliparity	19	43.2	21	67.7	**0.04**
Multiparity	25	56.8	10	32.3	
Not specified	1		3		
**Tobacco use**					
Yes	15	34.9	8	25.0	0.36
No	28	65.1	24	75.0	
Not specified	2		2		
**Gestational diabetes**					
Yes	2	4.5	6	19.4	**0.04**
No	42	95.5	25	80.6	
Not specified	1		3		
**Presence of HPV**					
Yes	10	22.2	8	23.5	0.89
No	35	77.8	26	76.5	
**Delivery route**					
Vaginal delivery	32	71.1	22	64.7	0.54
C-section	13	28.9	12	35.3	

Significant statistical values are marked with Bold.

**Table 4 jcm-10-05319-t004:** Univariate analysis of colposcopic factors by trimester associated with postpartum CIN regression.

	First Trimester					Second Trimester					Third Trimester					Postpartum Colposcopic Follow Up				
	Persistence of CIN *n* = 45		Regression of CIN *n* = 34			Persistence of CIN *n* = 45		Regression of CIN *n* = 34			Persistence of CIN *n* = 45		Regression of CIN *n* = 34			Persistence of CIN *n* = 45		Regression of CIN *n* = 34		
	*n*; (%) (Range)		*n*; (%) (Range)		*p*	*n*; (%) (Range)		*n*; (%) (Range)		*p*	*n*; (%) (Range)		*n*; (%) (Range)		*p*	*n*; (%) (Range)		*n*; (%) (Range)		*p*
Colposcopy																				
Yes	38	84.4	30	88.2	0.63	41	91.1	30	88.2	0.67	30	66.7	30	88.2	**0.03**	42	93.3	34	100	0.12
No	7	15.6	4	11.8		4	8.9	4	11.8		15	33.3	4	11.8		3	6.7	0	0	
Aspect																				
HSIL	11	64.7	10	66.7	0.51	25	71.4	16	61.5	0.21	22	78.6	12	54.5	0.1	25	80.6	11	45.8	**0.01**
LSIL	6	35.3	4	26.7		9	25.7	6	23.1		6	21.4	8	36.4		5	16.1	7	29.2	
Normal	0	0.0	1	6.7		1	2.9	4	15.4		0	0.0	2	9.1		1	3.2	6	25	
Not specified	21		15			6		4			2		8			11		10		
Cytology																				
HSIL	25	65.8	19	63.3	0.87	12	50.0	12	57.1	0.79	18	72.0	7	29.2	**0.009**	21	63.6	5	17.9	**0.003**
LSIL	7	18.4	7	23.3		8	33.3	5	23.8		4	16.0	12	50.0		3	9.1	7	25.0	
ASC-H	4	10.5	4	13.3		2	8.3	1	4.8		3	12.0	5	20.8		0	0	1	3.6	
Normal						2	8.3	3	14.3		5		6			9	27.3	15	53.6	
Not performed	2					17		9								9		6		
Biopsy																				
HSIL	8	88.9	11	91.7	0.83	14	87.5	3	60	0.39	7	77.8	2	50.0	**0.06**	25	78.1	0	0	**<0.0001**
LSIL	1	11.1	1	8.3		1	6.2	1	20		2	22.2	0	0		4	12.5	8	53.3	
Normal						1	6.2	1	20		0	0	2	50.0		3	9.4	7	46.7	
Not performed	29		18			25		25			21		26			10		19		

Significant statistical values are marked with Bold.

**Table 5 jcm-10-05319-t005:** Multivariate analysis of factors associated with postpartum CIN regression.

Variable	Odds Ratio (95% CI)	*p*
Parity status		
Multiparity	1	
Nulliparity	4.35 (1.03–18.42)	**0.046**
Gestational diabetes		
No	1	
Yes	6.00 (0.45–79.46)	0.17
Third trimester cervical cytology		
LSIL or ASC-H	1	
HSIL	0.17 (0.04–0.72)	**0.016**

Significant statistical values are marked with Bold.

## Data Availability

All data analyzed during this study are included in this article. Further enquiries can be directed to the corresponding author.

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
