# Peer review of "Predictive Factors Involved in Postpartum Regressions of Cytological/Histological Cervical High-Grade Dysplasia Diagnosed during Pregnancy"

_jcm, 2021, doi:10.3390/jcm10225319_

Round 1

Reviewer 1 Report

Table 1 is long and confusing.  I would recommend separating out findings according to trimester, it is hard to follow how it is presented now.  

There are comments about gestational diabetes being significant on univariate analysis.  The authors should comment about the low number of diabetics in their sample which may influence these findings.  There are also no other antepartum conditions evaluation (hypertension, etc) which I find interesting.

To really strengthen this study all women in the postpartum period would have biopsies, even if random.  This would increase the validity of confirming regression of CIN2+

I am confused by the language "postpartum control"  is this simply the follow up that every patient had?  Confusing as there is not really a control and an experimental group

Author Response

Response to Reviewer 1:

Please find attached the article containing the corrections and the tables. 

  • Table 1 is long and confusing.  I would recommend separating out findings according to trimester, it is hard to follow how it is presented now.  

We agree with you, and we modified this table as suggested

  • There are comments about gestational diabetes being significant on univariate analysis.  The authors should comment about the low number of diabetics in their sample which may influence these findings.  There are also no other antepartum conditions evaluation (hypertension, etc) which I find interesting.

We fully agree with you, and we believe that the low number of patients induced a loss of statistical power. We added the following precision in the comment part :

 « In our univariate analysis, there was a statistically significant result for the variable gestational diabetes, however this should be interpretated with caution due to the small patient cohort (8 patients) ».

About others conditions (such as HTA, infections during pregnancy or vaginal progesterone at the first trimester), this is a pertinent comment. These data were not collected due to the retrospective characteristic of the study and we focused on conditions that might have an impact on the immune system.

  • To really strengthen this study all women in the postpartum period would have biopsies, even if random.  This would increase the validity of confirming regression of CIN2+

Indeed, we agree with you and the data collected does not allow us to obtain such information, given the retrospective nature of the study. Notwithstanding this limitation, postpartum histological sampling was available in 71% of cases (56/79). This point has been modified in the result section: «71% (56/79) of our cohort had a histological diagnosis postpartum ».

  • I am confused by the language "postpartum control" is this simply the follow up that every patient had?  Confusing as there is not really a control and an experimental group

This is a pertinent comment, and the term “control” might be confusing. In this case, this term was substituted by "Postpartum colposcopic follow up".

Reviewer 2 Report

The conclusion paragraph needs more than just -- parity status may have an impact on dysplasia and then just talk about larger studies are needed.  Include info on recommendations for evaluation during pregnancy etc.

-Lines 51-52: This needs to be expanded on -- it is just one sentence.  i.e. why does postpartum regression occur?

-Why do we care if it regresses or not -- try to expand on that.

-Was this IRB approved?

-Expand on why third-trimester cytology may be associated with dysplasia regression?  Only parity is really discussed.

Author Response

Response to Reviewer 2

Please find attached the article containing the corrections and the tables

  • The conclusion paragraph needs more than just -- parity status may have an impact on dysplasia and then just talk about larger studies are needed.  Include info on recommendations for evaluation during pregnancy etc.

You’re right and we modified the conclusion as following: In conclusion, our study follows the ASCCP guidelines. Although assessment of high-grade dysplastic lesions may be postponed in the postpartum period, our observations revealed that the clinical value of surveillance for high-grade dysplasia during pregnancy is to identify lesions with a potential risk of progression to cervical carcinoma in the postpartum period. Our results highlighted that parity status may have an impact on the regression of dysplasia during pregnancy… »

  • Lines 51-52: This needs to be expanded on -- it is just one sentence.  i.e. why does postpartum regression occur?

Thank you for pointing this out, and this sentence deserves more information. We added this sentence:

« This regression is possibly related either to the return of the immune system or to a stimulation of it. Previous article suggested that during vaginal delivery, the cervical microlesions induced inflammation and consequently regression of the lesions. However, these results remain controversial ».

  • Why do we care if it regresses or not -- try to expand on that.

This is a relevant comment and, indeed the clinical benefit of knowing the evolution of high-grade dysplasia is to target patients who maintain persistent HSIL/CIN2+ throughout pregnancy and avoid loss to follow-up in the postpartum period, which is the key time to monitor persistence or the progression to a microinvasive lesion and treatment. If, on the other hand, regression is already noted prenatally and confirmed at postnatal follow-up, monitoring should be carried out at longer. We have added the following sentence to the beginning of the discussion section:

«The aim of this study was to assess the clinical interest in knowing which factors would allow targeting patients at potential risk of maintaining a high-grade lesion or progressing to a cancerous lesion in the postpartum period and avoiding loss to follow-up for adequate treatment. »

  • Was this IRB approved?

IRB was introduced according to the request for submission of the article in the section "institution review board statement" in line 235. We have also introduced it in the text in the methodology section. « This study was approved by the ethical research committee of the canton de Vaud (25.07.2019, ID N° 2017-01375)”.

  • Expand on why third-trimester cytology may be associated with dysplasia regression?  Only parity is really discussed.

You have raised an important point and here are some pieces of information that we will introduce in the discussion section :  « Persistent HPV infection is necessary for the development of precancers and cancers. In the third trimester, there is a ripening of the cervix due to hormonal change and consequently a return of the inflammatory system. The hypothesis is that this process allows clearance of the HPV infection and consequently a regression of the lesion. Cytology in the third trimester will allow this change to be observed and therefore predict the evolution of the lesions
